# Semi-Supervised Semantic Segmentation via Marginal Contextual Information

## Abstract

We present a novel confidence refinement scheme that enhances pseudo-labels in semi-supervised semantic segmentation. Unlike current leading methods, which filter pixels with low-confidence teacher predictions in isolation, our approach leverages the spatial correlation of labels in segmentation maps by grouping neighboring pixels and considering their pseudo-labels collectively. With this contextual information, our method, named S4MC, increases the amount of unlabeled data used during training while maintaining the quality of the pseudo-labels, all with negligible computational overhead. Through extensive experiments on standard benchmarks, we demonstrate that S4MC outperforms existing state-of-the-art semi-supervised learning approaches, offering a promising solution for reducing the cost of acquiring dense annotations. For example, S4MC achieves a substantial 6.34 mIoU improvement over the prior state-of-the-art method on PASCAL VOC 12 with 92 annotated images. The code to reproduce our experiments is available at https://s4mcontext.github.io/.

## 1 Introduction

Supervised learning has been the driving force behind advancements in modern computer vision, including classification (Krizhevsky et al., 2012; Dai et al., 2021), object detection (Girshick, 2015; Zong et al., 2022), and segmentation (Zagoruyko et al., 2016; Chen et al., 2018a; Li et al., 2022; Kirillov et al., 2023). However, it requires extensive amounts of labeled data, which can be costly and time-consuming to obtain. In many practical scenarios, while there is no shortage of available data, only a fraction can be labeled due to resource constraints. This challenge has led to the development of semi-supervised learning (SSL) (Rasmus et al., 2015; Berthelot et al., 2019b; Sohn et al., 2020a; Yang et al., 2022a), a methodology that leverages both labeled and unlabeled data for model training.

This paper focuses on applying SSL to semantic segmentation, which has applications in various areas such as perception for autonomous vehicles (Bartolomei et al., 2020), mapping (Van Etten et al., 2018) and agriculture (Milioto et al., 2018). SSL is particularly appealing for segmentation tasks, as manual labeling can be prohibitively expensive.

A widely adopted approach for SSL is pseudo-labeling (Lee, 2013; Arazo et al., 2020). This technique dynamically assigns supervision targets to unlabeled data during training based on the model's predictions. To generate a meaningful training signal, it is essential to adapt the predictions before integrating them into the learning process. Several techniques have been proposed for that, such as using a teacher network to generate supervision to a student network (Hinton et al., 2015). The teacher network can be made more powerful during training by applying a moving average to the student network's weights (Tarvainen and Valpola, 2017). Additionally, the teacher may undergo weaker augmentations than the student (Berthelot et al., 2019b), simplifying the teacher's task.

Submitted to 37th Conference on Neural Information Processing Systems (NeurIPS 2023). Do not distribute.

**Before refinement**  **After refinement**

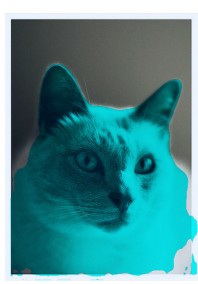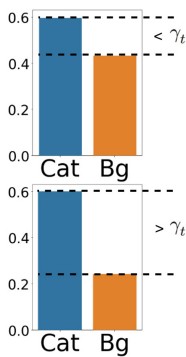

Figure 1: **Confidence refinement. Left:** pseudo-labels generated by the teacher network without refinement. **Middle:** pseudo-labels obtained from the same model after refinement with marginal contextual information. **Right Top:** predicted probabilities of the top two classes of the pixel highlighted by the red square before, and **Bottom:** after refinement. S4MC allows additional correct pseudo labels to propagate.

However, pseudo-labeling is intrinsically susceptible to confirmation bias, which tends to reinforce the model predictions instead of improving the student model. Mitigating confirmation bias becomes particularly important when dealing with erroneous predictions made by the teacher network.

One popular technique to address this issue is confidence-based filtering (Sohn et al., 2020a). This approach assigns pseudo-labels only when the model's confidence surpasses a specified threshold, thereby reducing the number of incorrect pseudo-labels. Though simple, this strategy was proven effective and inspired multiple improvements in semi-supervised classification (Zhang et al., 2021; Rizve et al., 2021), segmentation (Wang et al., 2022), and object detection in images (Sohn et al., 2020b; Liu et al., 2021) and 3D scenes (Zhao et al., 2020; Wang et al., 2021). However, the strict filtering of the supervision signal leads to extended training periods and, potentially, to overfitting when the labeled instances used are insufficient to represent the entire sample distribution. Lowering the threshold would allow for higher training volumes at the cost of reduced quality, further hindering the performance (Sohn et al., 2020a).

In response to these challenges, we introduce a novel confidence refinement scheme for the teacher network predictions in segmentation tasks, designed to increase the availability of pseudo-labels without sacrificing their accuracy. Drawing on the observation that labels in segmentation maps exhibit strong spatial correlation, we propose to group neighboring pixels and collectively consider their pseudo-labels. When considering pixels in spatial groups, we asses the event-union probability, which is the probability that at least one pixel belongs to a given class. We assign a pseudo-label if this probability is sufficiently larger that the event-union probability of any other class. By taking context into account, our approach *Semi-Supervised Semantic Segmentation via Marginal Contextual Information* (S4MC), enables a relaxed filtering criterion which increases the number of unlabeled pixels utilized for learning while maintaining high-quality labeling, as demonstrated in Fig. 1.

We evaluated S4MC on multiple semi-supervised segmentation benchmarks. S4MC achieves significant improvements in performance over previous state-of-the-art methods. In particular, we observed a remarkable increase of **+6.34 mIoU** on PASCAL VOC 12 (Everingham et al., 2010) using only 92 annotated images and an increase of **+1.85 mIoU** on Cityscapes (Cordts et al., 2016) using only 186 annotated images. These findings highlight the effectiveness of S4MC in producing high-quality segmentation results with minimal labeled data.

## 2 Related Work

### 2.1 Semi-Supervised Learning

Pseudo-labeling (Lee, 2013) is a popular and effective technique in SSL, where labels are assigned to unlabeled data based on model predictions. To make the most of these labels during training, it is essential to refine them (Laine and Aila, 2016; Berthelot et al., 2019b,a; Xie et al., 2020). One way to achieve this is through consistency regularization (Laine and Aila, 2016; Tarvainen and Valpola, 2017; Miyato et al., 2018), which ensures consistent predictions between different views of the unlabeled

data. Alternatively, a teacher model can be used to obtain pseudo-labels, which are then used to train a student model. To ensure that the pseudo-labels are useful, the temperature of the prediction (soft pseudo-labels; Berthelot et al., 2019b) can be increased, or the label can be assigned to samples with high confidence (hard pseudo-labels; Xie et al., 2020; Sohn et al., 2020a; Zhang et al., 2021). In this work we follow the hard pseudo-label assignment approach and improve upon previous methods by proposing a confidence refinement scheme.

## 2.2 Semi-Supervised Semantic Segmentation

In semantic segmentation, most SSL methods rely on a combination of consistency regularization and the development of augmentation strategies compatible with segmentation tasks. Given the uneven distribution of labels typically encountered in segmentation maps, techniques such as adaptive sampling, augmentation, and loss re-weighting are commonly employed (Hu et al., 2021). Feature perturbations on unlabeled data (Ouali et al., 2020; Zou et al., 2021; Liu et al., 2022b) are also used to enhance consistency, along with the application of virtual adversarial training (Liu et al., 2022b). Curriculum learning strategies that incrementally increase the proportion of data used over time are beneficial in exploiting more unlabeled data (Yang et al., 2022b; Wang et al., 2022). A recent approach introduced by (Wang et al., 2022) cleverly utilizes *unreliable* predictions by employing contrastive loss with the least confident classes predicted by the model. However, most existing works primarily focus on individual pixel label predictions. In contrast, we delve into the contextual information offered by spatial predictions on unlabeled data.

## 2.3 Contextual Information

Contextual information encompasses environmental cues that assist in interpreting and extracting meaningful insights from visual perception (Toussaint, 1978; Elliman and Lancaster, 1990). Incorporating spatial context explicitly has been proven beneficial in segmentation tasks, for example, by encouraging smoothness like in the Conditional Random Fields (CRF) method (Chen et al., 2018a) and attention mechanisms (Vaswani et al., 2017; Dosovitskiy et al., 2021; Wang et al., 2020). Combating dependence on context has shown to be useful by Nekrasov et al. (2021). In this work, we leverage the context from neighboring pixel predictions to enhance pseudo-label propagation.

# 3 Method

## 3.1 Overview

In semi-supervised semantic segmentation, we are given a labeled training set of images $\mathcal{D}_\ell = \left\{(\mathbf{x}_i^\ell, \mathbf{y}_i)\right\}_{i=1}^{N_\ell}$, and an unlabeled set $\mathcal{D}_u = \{\mathbf{x}_i^u\}_{i=1}^{N_u}$ sampled from the same distribution, i.e., $\left\{\mathbf{x}_i^\ell, \mathbf{x}_i^u\right\} \sim D_x$. Here, $\mathbf{y}$ are 2D tensors of shape $H \times W$, assigning a semantic label to each pixel of $\mathbf{x}$. We aim to train a neural network $f_\theta$ to predict the semantic segmentation of unseen images sampled from $D_x$.

We follow a teacher–student approach (Tarvainen and Valpola, 2017) and train two networks $f_{\theta_s}$ and $f_{\theta_t}$ that share the same architecture but update their parameters separately. The student network $f_{\theta_s}$ is trained using supervision from the labeled samples and pseudo-labels created by the teacher's predictions for unlabeled ones. The teacher model $f_{\theta_t}$ is updated as an exponential moving average (EMA) of the student weights. $f_{\theta_s}(\mathbf{x}_i)$ and $f_{\theta_t}(\mathbf{x}_i)$ denote the predictions of the student and teacher models for the $\mathbf{x}_i$ sample, respectively. At each training step, a batch of $\mathcal{B}_\ell$ and $\mathcal{B}_u$ images is sampled from $\mathcal{D}_\ell$ and $\mathcal{D}_u$, respectively. The optimization objective can be written as the following loss:

$$\mathcal{L} = \mathcal{L}_s + \lambda \mathcal{L}_u \tag{1}$$

$$\mathcal{L}_s = \frac{1}{M_l} \sum_{\mathbf{x}_i^\ell, \mathbf{y}_i \in \mathcal{B}_l} \ell_{CE}(f_{\theta_s}(\mathbf{x}_i^\ell), \mathbf{y}_i) \tag{2}$$

$$\mathcal{L}_u = \frac{1}{M_u} \sum_{\mathbf{x}_i^u \in \mathcal{B}_u} \ell_{CE}(f_{\theta_s}(\mathbf{x}_i^u), \hat{\mathbf{y}}_i), \tag{3}$$

where $\mathcal{L}_s$ and $\mathcal{L}_u$ are the losses over the labeled and unlabeled data correspondingly, $\lambda$ is a hyperparameter controlling their relative weight, and $\hat{\mathbf{y}}_i$ is the pseudo-label for the $i$-th unlabeled

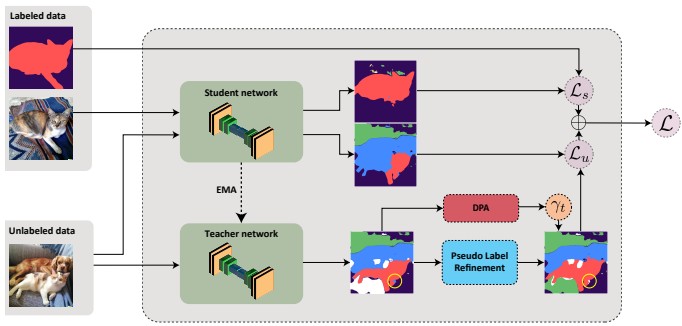
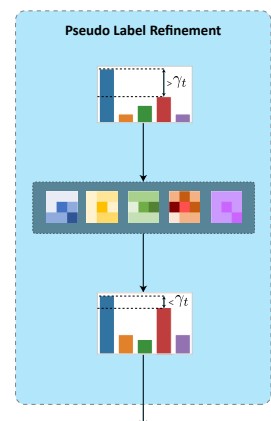

Figure 2: **Left:** S4MC employs a teacher-student paradigm for semi-supervised segmentation. Labeled images are used to supervise the student network directly. Both teacher and student networks process unlabeled images. Predictions from the teacher network are refined and used to evaluate the margin value, which is then thresholded to produce pseudo-labels that guide the student network. The threshold, denoted as $\gamma_t$, is dynamically adjusted based on the teacher network's predictions. **Right:** Our confidence refinement module exploits neighboring pixels to adjust per-class predictions, as detailed in Section 3.2.1. The class distribution of the pixel marked by the yellow circle on the left is changed. Before refinement, the margin surpasses the threshold and erroneously assigns the blue class (dog) as a pseudo-label. However, after refinement, the margin significantly reduces, thereby preventing the propagation of this error.

image. Not every pixel of $\mathbf{x}_i$ has a corresponding label or pseudo-label, and $M_l$ and $M_u$ denote the number of pixels with label and assigned pseudo-label in the image batch, respectively.

### 3.1.1 Pseudo-label Propagation

For a given image $\mathbf{x}_i$, we denote by $\mathbf{x}_{j,k}^i$ the pixel in the $j$-th row and $k$-th column. We adopt a thresholding-based criterion inspired by (Sohn et al., 2020a). By establishing a score, denoted as $\kappa$, which is based on the class distribution predicted by the teacher network, we assign a pseudo-label to a pixel if its score exceeds a threshold $\gamma_t$:

$$\hat{\mathbf{y}}_{j,k}^i = \begin{cases} \arg\max_c \{p_c(x_{j,k}^i)\} & \text{if } \kappa(x_{j,k}^i; \theta_t) > \gamma_t, \\ \text{ignore} & \text{otherwise,} \end{cases} \tag{4}$$

where $p_c(x_{j,k}^i)$ is the pixel probability of class $c$. A commonly used score is given by $\kappa(x_{j,k}^i; \theta_t) = \max_c \{p_c(x_{j,k}^i)\}$. However, we found that using a pixel-wise margin, inspired by the work of Scheffer et al. (2001) and Shin et al. (2021), produces more stable results. This approach calculates the margin as the difference between the highest and the second-highest values of the probability vector:

$$\kappa_{\text{margin}}(x_{j,k}^i) = \max_c \{p_c(x_{j,k}^i)\} - \max_c 2\{p_c(x_{j,k}^i)\}. \tag{5}$$

### 3.1.2 Dynamic Partition Adjustment (DPA)

Following U²PL (Wang et al., 2022), we use a decaying threshold $\gamma_t$. DPA replaces the fixed threshold with a quantile-based threshold that decreases with time. At each iteration, we set $\gamma_t$ as the $\alpha_t$-th quantile of $\kappa_{\text{margin}}$ over all pixels of all images in the batch. $\alpha_t$ is defined as follows:

$$\alpha_t = \alpha_0 \cdot (1 - {}^t/\text{iterations}). \tag{6}$$

As the model predictions improve with each iteration, gradually lowering the threshold increases the number of propagated pseudo-labels without compromising their quality.

### 3.2 Marginal Contextual Information

Utilizing contextual information (Section 2.3), we look at surrounding predictions (predictions on neighboring pixels) to refine the semantic map at each pixel. We introduce the concept of "Marginal

Contextual Information," which involves integrating additional information to enhance predictions across all classes. At the same time, reliability-based pseudo-label methods focus on the dominant class only (Sohn et al., 2020a; Wang et al., 2023). Section 3.2.1 describes our confidence refinement, followed by our thresholding strategy and a description of S4MC methodology.

### 3.2.1 Confidence Margin Refinement

We refine the predicted pseudo-label of each pixel by considering the predictions of its neighboring pixels. Given a pixel $x_{j,k}^i$ with a corresponding per-class prediction $p_c(x_{j,k}^i)$, we examine neighboring pixels $x_{\ell,m}^i$ within an $N \times N$ pixel neighborhood surrounding it. We then calculate the probability that at least one of the two pixels belongs to class $c$:

$$\tilde{p}_c(x_{j,k}^i) = p_c(x_{j,k}^i) + p_c(x_{\ell,m}^i) - p_c(x_{j,k}^i, x_{\ell,m}^i), \tag{7}$$

where $p_c(x_{j,k}^i, x_{\ell,m}^i)$ denote the joint probability of both $x_{j,k}^i$ and $x_{\ell,m}^i$ belonging to the same class $c$.

While the model does not predict joint probabilities, it is reasonable to assume a non-negative correlation between the probabilities of neighboring pixels. This is largely due to the nature of segmentation maps, which are typically piecewise constant. Consequently, any information regarding the model's prediction of neighboring pixels belonging to a specific class should not lead to a reduction in the posterior probability of the given pixel also falling into that class. The joint probability can thus be bounded from below by assuming independence: $p_c(x_{j,k}^i, x_{\ell,m}^i) \geqslant p_c(x_{j,k}^i) \cdot p_c(x_{\ell,m}^i)$. By substituting this into Eq. (7), we obtain an upper bound for the event union probability:

$$\tilde{p}_c(x_{j,k}^i) \leq p_c(x_{j,k}^i) + p_c(x_{\ell,m}^i) - p_c(x_{j,k}^i) \cdot p_c(x_{\ell,m}^i). \tag{8}$$

This formulation allows us to filter out confidence margins that do not exceed the threshold.

For each class $c$, we select the neighbor with the maximal information gain using Eq. (8):

$$\tilde{p}_c^{\mathbf{N}}(x_{j,k}^i) = \max_{\ell,m} \tilde{p}_c(x_{j,k}^i). \tag{9}$$

Computing the event union over all classes employs neighboring predictions to amplify differences in ambiguous cases. Consider, for instance, an uncertain pixel prediction with a $0.5$ probability of belonging to one of two classes. If a neighboring pixel has a $0.7$ probability of belonging to the first class and only a $0.3$ probability of belonging to the second, this results in a significant event union probabilities margin of $0.2$. Similarly, this prediction refinement prevents the creation of over-confident predictions that is not supported by additional spatial evidence and helps in reducing confirmation bias. The refinement is visualized in Fig. 1. In our experiments, we used a neighborhood size of $3 \times 3$. To determine whether the incorporation of contextual information could be enhanced with larger neighborhoods, we conducted an ablation study focusing on the neighborhood size and the neighbor selection criterion, as detailed in Table 4a. For larger neighborhoods, we decrease the probability contribution of the neighboring pixels with a distance-dependent factor:

$$\tilde{p}_c(x_{j,k}^i) = p_c(x_{j,k}^i) + \beta_{\ell,m} \left[ p_c(x_{\ell,m}^i) - p_c(x_{j,k}^i, x_{\ell,m}^i) \right], \tag{10}$$

where $\beta_{\ell,m} = \exp\left(-\frac{1}{2}(|\ell - j| + |m - k|)\right)$ is a spatial weighting function. Empirically, contextual information refinement affects mainly the most probable one or two classes. This aligns well with our choice to use the margin confidence (5).

Considering more than two events (more than one neighbor), one can use the formulation for three or four event-union. In practice, we find two event-union using Eq. (10), assign it as $p_c(x_{j,k}^i)$, find the next desired event using Eq. (9) with the remaining neighbors, and repeat the process.

### 3.2.2 Threshold Setting

Setting a high threshold can mitigate confirmation bias from the teacher model's "beliefs" transferring to the student model. However, this comes at the expense of learning from fewer examples, potentially resulting in a less comprehensive model. *Dynamic Partition Adjustment* (DPA; Wang et al., 2022) attempt to address this issue by setting a threshold that decreases over time. We adopt this method in determining the threshold from the teacher predictions pre-refinement $p_c(x_{j,k}^i)$, but we filter values based on $\tilde{p}_c(x_{j,k}^i)$. Consequently, more pixels pass the threshold that remains unaffected. We set $\alpha_0 = 0.4$, i.e., 60% of raw predictions pass the threshold at $t = 0$, as this value demonstrated superior performance in our experiments. An ablation study for $\alpha_0$ is provided in Table 4b.

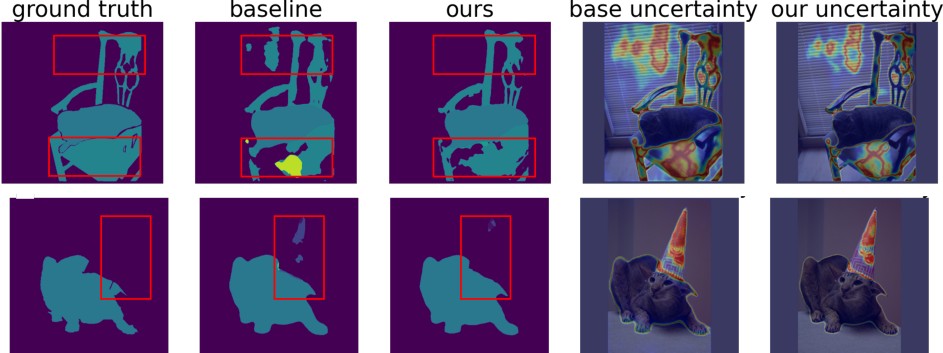

Figure 3: **Qualitative results of S4MC.** The outputs of two trained models and the annotated ground truth. The segmentation map predicted by S4MC (*Ours*) compared to the segmentation map using no refinement module (*Baseline*) and to the ground truth. *Heat map* represents the uncertainty of the model as $\kappa^{-1}$, showing a more confident prediction over certain areas, yielding to a smother segmentation maps (compared in the red boxes).

### 3.3 Putting it All Together

We perform semi-supervised for semantic segmentation by pseudo-labeling pixels using their neighbors' contextual information. Labeled images are only fed into the student model, producing the supervised loss (Eq. (2)). Unlabeled images are fed into the student and teacher models. We sort the margin based $\kappa_{\mathrm{margin}}$ (Eq. (5)) values of teacher predictions and set $\gamma_t$ as described in Section 3.2.2. The per-class teacher predictions are refined using the *weighted union event* relaxation, as defined in Eq. (10). Pixels with higher margin values than $\gamma_t$ are assigned with pseudo-labels as described in Eq. (4), producing the unsupervised loss (Eq. (3)). A visualization of the entire pipeline is depicted in Fig. 2.

The impact of S4MC is demonstrated in Fig. 4, which compares the fraction of pixels that pass the threshold with and without refinement. (a) Our method makes greater use of unlabeled data during most of the training process, (b) while the refinement ensures high-quality pseudo-labels. Qualitative results are presented in Fig. 3, where one can see both the confidence heatmap and the pseudo-labels with and without the impact of S4MC.

## 4 Experiments

This section presents our experimental results. The setup for the different datasets and partition protocols is detailed in Section 4.1. Section 4.2 compares our method against existing approaches and Section 4.3 provides the ablation study. Further implementation details are given in the Appendix.

### 4.1 Setup

**Datasets** In our experiments, we use PASCAL VOC 2012 (Everingham et al., 2010) and Cityscapes (Cordts et al., 2016) datasets.

The PASCAL VOC dataset comprises 20 object classes (plus background). The dataset includes 2,913 annotated images, divided into a training set of 1,464 images and a validation set of 1,449 images. In addition, the dataset includes 9,118 coarsely annotated training images (Hariharan et al., 2011), in which only a subset of the pixels are labeled. Following previous research, we conduct two sets of experiments. The 'classic' experiment utilizes only the original training set (Wang et al., 2022; Zou et al., 2021), while the 'coarse' experiment uses all available data (Wang et al., 2022; Chen et al., 2021; Hu et al., 2021).

The Cityscapes (Cordts et al., 2016) dataset includes urban scenes from 50 different cities with 30 classes, of which only 19 are typically used for evaluation (Chen et al., 2018a,b). Similarly to PASCAL, in addition to 2,975 training and 500 validation images, the dataset includes 19,998 coarsely annotated images, which we do not use in our experiment.

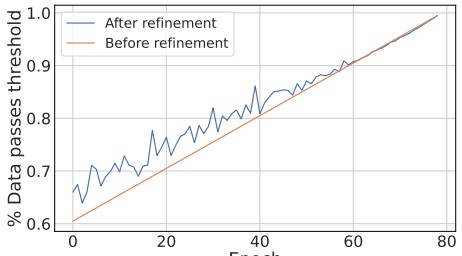
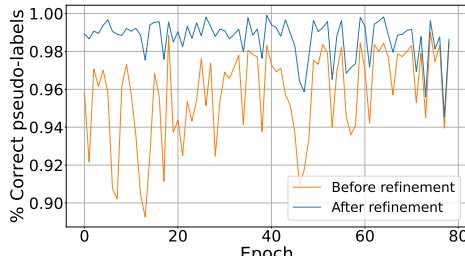

(a) **Data fraction that passes the threshold**. The baseline model has a fixed percentage, as it is based on DPA. Our method increases the number of pixels assigned pseudo-label, mostly in the early stage of the training when the model is under-confident.

(b) **Fraction of correct pseudo-labels** the assigned pseudo-labels with the correct class divided by the total assigned pseudo-label. S4MC produces more quality pseudo-labels during the training process, most notably at the early stages.

Figure 4: Pseudo-label quantity and quality on PASCAL VOC 2012 (Everingham et al., 2010) with 366 labeled images using our margin (5) confidence function.

Table 1: Comparison between our method and prior art on the PASCAL VOC 2012 `val` on different partition protocols. the caption describes the share of the training set used as labeled data and, in parentheses, the actual number of labeled images. Larger improvement can be observed for partitions of extremely low annotated data, where other methods suffer from starvation due to poor teacher generalization.

| Method | 1/16 (92) | 1/8 (183) | 1/4 (366) | 1/2 (732) | Full (1464) |
|---|---|---|---|---|---|
| Supervised Only | 45.77 | 54.92 | 65.88 | 71.69 | 72.50 |
| CutMix-Seg (French et al., 2020) | 52.16 | 63.47 | 69.46 | 73.73 | 76.54 |
| PseudoSeg (Zou et al., 2021) | 57.60 | 65.50 | 69.14 | 72.41 | 73.23 |
| PC$^2$Seg (Zhong et al., 2021) | 57.00 | 66.28 | 69.78 | 73.05 | 74.15 |
| CPS (Chen et al., 2021) | 64.10 | 67.40 | 71.70 | 75.90 | - |
| ReCo (Liu et al., 2022a) | 64.80 | 72.0 | 73.10 | 74.70 | - |
| ST++ (Yang et al., 2022b) | 65.2 | 71.0 | 74.6 | 77.3 | 79.1 |
| U$^2$PL (Wang et al., 2022) | 67.98 | 69.15 | 73.66 | 76.16 | 79.49 |
| PS-MT (Liu et al., 2022b) | 65.8 | 69.6 | 76.6 | 78.4 | 80.0 |
| S4MC + CutMix-Seg (Ours) | 70.96 | 71.69 | 75.41 | 77.73 | 80.58 |
| S4MC + FixMatch (Ours) | **74.32** | **75.62** | **77.84** | **79.72** | **81.51** |

**Implementation details** We implement S4MC on top of two framework variants: CutMix-Seg (French et al., 2020) and FixMatch (Sohn et al., 2020a). Both use DeepLabv3+ (Chen et al., 2018b) with a Imagenet-pre-trained (Russakovsky et al., 2015) ResNet-101 (He et al., 2016). The teacher parameters $\theta_t$ are updated via an exponential moving average (EMA) of the student parameters Tarvainen and Valpola (2017): $\theta_t^\eta = \tau\theta_t^{\eta-1} + (1-\tau)\theta_s^\eta$, where $0 \leq \tau \leq 1$ defines how close the teacher is to the student and $\eta$ denotes the training iteration. We used $\tau = 0.99$. Additional details are provided in Appendix D.

**Evaluation** We compare S4MC with baselines under the common partition protocols – using $1/2$, $1/4$, $1/8$, and $1/16$ of the training data as labeled data. For the 'classic' setting of the PASCAL experiment, we additionally compare using all the finely annotated images. We follow standard protocols and use mean Intersection over Union (mIoU) as our evaluation metric. We use the data split published by Wang et al. (2022) when available to ensure a fair comparisons. For the ablation studies, we use PASCAL VOC 2012 `val` with $1/4$ partition.

**Methods in comparison** We compare against popular SSL segmentation methods: CutMix-Seg (French et al., 2020), CCT (Ouali et al., 2020), GCT (Ke et al., 2020), PseudoSeg (Zou et al., 2021), CPS (Chen et al., 2021), PC$^2$Seg (Zhong et al., 2021), AEL (Hu et al., 2021), U$^2$PL (Wang et al.,

Table 2: Comparison between our method and prior art on the 'coarse' PASCAL VOC 2012 `val` dataset under different partition protocols, using additional unlabeled data from (Hariharan et al., 2011). For each partition ratio we included the number of labeled images in parentheses. As in 1, larger improvements are observed for partitions with less annotated data.

| Method | 1/16 (662) | 1/8 (1323) | 1/4 (2646) | 1/2 (5291) |
|---|---|---|---|---|
| Supervised Only | 67.87 | 71.55 | 75.80 | 77.13 |
| CutMix-Seg (French et al., 2020) | 71.66 | 75.51 | 77.33 | 78.21 |
| CCT (Ouali et al., 2020) | 71.86 | 73.68 | 76.51 | 77.40 |
| GCT (Ke et al., 2020) | 70.90 | 73.29 | 76.66 | 77.98 |
| CPS (Chen et al., 2021) | 74.48 | 76.44 | 77.68 | 78.64 |
| AEL (Hu et al., 2021) | 77.20 | 77.57 | 78.06 | 80.29 |
| PS-MT (Liu et al., 2022b) | 75.5 | 78.2 | 78.7 | - |
| U$^2$PL (Wang et al., 2022) | 77.21 | 79.01 | 79.3 | 80.50 |
| S4MC + CutMix-Seg (Ours) | 78.49 | 79.67 | 79.85 | 81.11 |
| S4MC + FixMatch (Ours) | **80.77** | **81.9** | **82.3** | **83.3** |

Table 3: Comparison between our method and prior art on the Cityscapes `val` dataset under different partition protocols. Labeled and unlabeled images are selected from the Cityscapes `training` dataset. For each partition protocol, the caption gives the share of the training set used as labeled data, in parentheses, the number of labeled images.

| Method | 1/16 (186) | 1/8 (372) | 1/4 (744) | 1/2 (1488) |
|---|---|---|---|---|
| Supervised Only | 62.96 | 69.81 | 74.08 | 77.46 |
| CutMix-Seg (French et al., 2020) | 69.03 | 72.06 | 74.20 | 78.15 |
| CCT (Ouali et al., 2020) | 69.32 | 74.12 | 75.99 | 78.10 |
| GCT (Ke et al., 2020) | 66.75 | 72.66 | 76.11 | 78.34 |
| CPS (Chen et al., 2021) | 69.78 | 74.31 | 74.58 | 76.81 |
| AEL (Hu et al., 2021) | 74.45 | 75.55 | 77.48 | 79.01 |
| U$^2$PL (Wang et al., 2022) | 70.30 | 74.37 | 76.47 | 79.05 |
| PS-MT (Liu et al., 2022b) | - | 76.89 | 77.6 | 79.09 |
| S4MC + CutMix-Seg (Ours) | 75.03 | 77.02 | 78.78 | 78.86 |
| S4MC + FixMatch (Ours) | **76.3** | **78.25** | **78.95** | **79.13** |

2022), PS-MT (Liu et al., 2022b), and ST++ (Yang et al., 2022b). "Supervised Only" stands for supervised training without using any unlabeled data. As a baseline, we use CutMix-Seg (French et al., 2020).

## 4.2 Results

**PASCAL VOC 2012.** Table 1 compares our method with state-of-the-art baselines on the PASCAL VOC 2012 dataset. While Table 2 shows the comparison results on the PASCAL VOC 2012 dataset with additional coarsely annotated data from SBD (Hariharan et al., 2011). In both setups, S4MC outperform all the compared methods in standard partition protocols, both when using labels only for the original PASCAL VOC 12 dataset and when using SBD annotations as well. Qualitative results are shown in Fig. 3. As can be seen our refinement procedure aids in both adding falsely filtered pseduo-labels as well as removing erroneous ones.

**Cityscapes.** Table 3 Presents the comparison results on the Cityscapes `val` dataset. Table 3 compares our method with other state-of-the-art methods on the Cityscapes (Cordts et al., 2016) dataset under various partition protocols. S4MC outperforms the compared methods in most partitions, except for the 1/2 setting, and combined with Fixmatch scheme, S4MC outperforms compared approaches across all partitions.

Table 4: The effect of neighborhood size and neighbor selection criterion.

(a) Neighborhood choice.

| Selection criterion | Neighborhood size N | | |
|---|---|---|---|
| | $3 \times 3$ | $5 \times 5$ | $7 \times 7$ |
| Random neighbor | 73.25 | 71.1 | 70.41 |
| Max neighbor | **75.41** | 75.18 | 74.89 |
| Min neighbor | 74.54 | 74.11 | 70.28 |
| Two max neighbors | 74.14 | 75.15 | 74.36 |

(b) $\alpha_0$ in Eq. (6), which controls the initial proportion of confidence pixels

| 20% | 30% | 40% | 50% | 60% |
|---|---|---|---|---|
| 74.45 | 73.85 | **75.41** | 74.56 | 74.31 |

**Contextual information at inference.** Given that our margin refinement scheme operates through prediction adjustments, we explored whether it could be employed at inference time to further enhance performance. The results reveal a negligible improvement in the DeepLab-V3-plus model, from an 85.7 mIOU to 85.71. This underlines that the performance advantage of S4MC primarily derives from the adjusted margin, as the most confident class is rarely swapped. A heatmap of the prediction over several samples is presented in Fig. 3 and Appendix E.

## 4.3 Ablation Study

We ablate different components of our method using the CutMix-Seg framework variant, and evaluated using the Pascal VOC 12 dataset with a partition protocol of $1/4$ labeled images.

**Neighborhood size and neighbor selection criterion.** Our prediction refinement scheme employs event-union probability with neighboring pixels, which depends on the chosen neighbor to pair with the current pixel. To assess this, we tested varying neighborhood sizes ($N = 3, 5, 7$) and criteria for selecting the neighboring pixel: (a) random, (b) maximal class probability, (c) minimal class probability, and (d) two neighbors, as described in Section 3.2.1. As shown in Table 4a, a small $3 \times 3$ neighborhood with one neighboring pixel of the highest class probability proved most efficient in our experiments.

**Threshold parameter tuning** As outlined in Section 3.1.2, we utilize a dynamic threshold that depends on an initial value, $\alpha_0$. In Table 4b, we examine the effect of different initial quantiles to establish this threshold. A smaller $\alpha_0$ would propagate too many errors, leading to significant confirmation bias. In contrast, a larger $\alpha_0$ would mask most of the data, resulting in insufficient label propagation, rendering the semi-supervised learning process lengthy and inefficient. We found that an $\alpha_0$ of 40% yields the best performance.

## 5 Conclusion

In this paper, we introduce S4MC, a novel approach for incorporating spatial contextual information in semi-supervised segmentation. This strategy refines confidence levels and enables us to leverage a larger portion of unlabeled data. S4MC outperforms existing approaches and achieves state-of-the-art results on multiple popular benchmarks under various data partition protocols, such as Cityscapes and Pascal VOC 12. While we believe S4MC offers a good solution to lowering the annotation requirement, it has several limitations. First, the event-union relaxation is relevant in problems where spatial coherency is expected. The generalization of our framework to other dense prediction tasks would necessitate an assessment of whether this relaxation is applicable. Furthermore, our method employs a fixed shape neighborhood without considering the structure of objects. It would be intriguing to investigate the use of segmented regions to define new neighborhoods, and this is a direction we plan to explore in the future.

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
