## A  Pseudo-labels quality analysis

The quality improvement and the quantity increase of pseudo-labels are shown in Fig. 4. Further analysis of the quality improvement of our method is demonstrated in Fig. A.1 by separating the *true positive* and *false positive*.

Within the initial phase of the learning process, the enhancement in the quality of pseudo-labels can be primarily attributed to the advancement in true positive labels. In our method, the refinement not only facilitates the inclusion of a larger number of pixels surpassing the threshold but also ensures that a significant majority of these pixels are of high quality.

As the learning process progresses, most improvements are obtain from a decrease in false positives pseudo-labels. This analysis shows that our method effectively minimizes the occurrence of incorrect pseudo-labeled, particularly when the threshold is set to a lower value. In other words, our method reduce conformation bias that comes from the decaying of the threshold as the learning process progress.

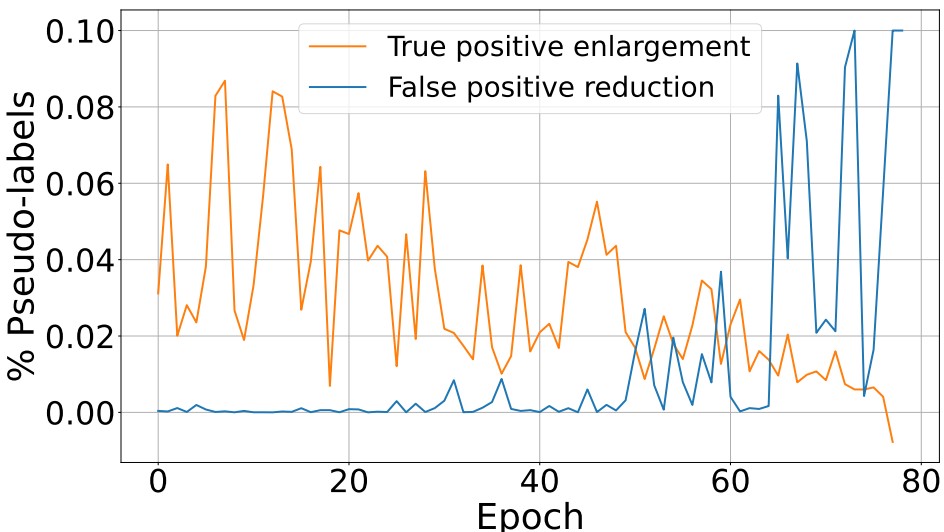

Figure A.1: **Quality of pseudo-labels**, on PASCAL VOC 2012 (Everingham et al., 2010) over training iterations. Fig. 4 separated to *True positive* and *False positive* analysis. *True positive* are the bigger part of improvement at early stage of the training process, while reduction of *false positive* is the main contribution late in the training process

## B  Confidence function alternatives

In this paper, we introduce a confidence function to determine pseudo-label propagation. We introduced $\kappa_{\mathrm{margin}}(x_{i,j})$ and mentioned other alternatives have been examined.

Here we define several options for the confidence function.

The simplest option is to look at the probability of the dominant class,

$$\kappa_{\max}(x_{j,k}^i) = \max_c p_c(x_{j,k}^i),  \tag{B.1}$$

which is commonly used to generate pseudo-labels.

The second alternative is negative entropy, defined as

$$\kappa_{\mathrm{ent}}(x_{j,k}^i) = \sum_{c \in C} p_c(x_{j,k}^i) \log\big(p_{i,j}^c\big).  \tag{B.2}$$

Table B.1: Ablation study on the confidence function $\kappa$, over Pascal VOC 12 with partition protocols

| Function | 1/4 (366) | 1/2 (732) | Full (1464) |
|---|---|---|---|
| $\kappa_{\text{max}}$ | 74.29 | 76.16 | 79.49 |
| $\kappa_{\text{ent}}$ | 75.18 | 77.55 | 79.89 |
| $\kappa_{\text{margin}}$ | 75.41 | 77.73 | 80.58 |

Note that this is indeed a confidence function since high entropy corresponds to high uncertainty, and low entropy corresponds to high confidence.

The third option is for us to define the margin function (Scheffer et al., 2001; Shin et al., 2021) as the difference between the first and second maximal values of the probability vector and also described in the main paper:

$$\kappa_{\text{margin}}(x_{i,j}) = \max_c(p_c(x_{j,k}^i)) - \text{max2}_c(p_c(x_{j,k}^i)), \tag{B.3}$$

where $\text{max2}$ denotes the vector's second maximum value. All alternatives are compared in Table B.1.

Table B.1 studies the impact of different confidence functions on pseudo-label refinement. We found that using a margin to describe confidence is a suitable way when there is a contradiction in smooth regions.

## C  Bounding the joint probability

In this paper, we had the union event estimation with the independence assumption, defined as

$$p_c^1(x_{j,k}^i, x_{\ell,m}^i) \approx p_c(x_{j,k}^i) \cdot p_c(x_{\ell,m}^i) \tag{C.1}$$

In addition to the independence approximation, it is possible to estimate the unconditional expectation of two neighboring pixels belonging to the same class based on labeled data:

$$p_c^2(x_{j,k}^i, x_{\ell,m}^i) = \frac{1}{|\mathcal{N}_l| \cdot H \cdot W \cdot |\mathbf{N}|} \sum_{i \in \mathcal{N}_l} \sum_{j,k \in H \times W} \sum_{\ell,m \in \mathbf{N}_{j,k}} \mathbb{1}\{y_{j,k}^i = y_{\ell,m}^i\}. \tag{C.2}$$

To avoid overestimating that could lead to overconfidence, we set

$$p_c(x_{j,k}^i, x_{\ell,m}^i) = \max(p_c^1(x_{j,k}^i, x_{\ell,m}^i), p_c^2(x_{j,k}^i, x_{\ell,m}^i)) \tag{C.3}$$

That upper bound of joint probability ensures that the independence assumption does not underestimate the joint probability, preventing overestimating the union event probability. Using Eq. (C.3) increase the mIOU by **0.22** on average, compared to non use of S4MC refinement, using 366 annotated images from PASCAL VOC 12 Using only Eq. (C.2) reduced the mIOU by **-14.11** compared to non use of S4MC refinement and actually harmed the model capabilities to produce quality pseudo-labels.

## D  Implementation Details

All experiments were conducted for 80 training epochs with the simple stochastic gradient descent (SGD) optimizer with a momentum of 0.9 and learning rate policy of $lr = lr_{\text{base}} \cdot \left(1 - \frac{\text{iter}}{\text{total iter}}\right)^{\text{power}}$. With the probability of 0.5, we apply CutMix (Yun et al., 2019) augmentation on the unlabeled data.

For PASCAL VOC 2012 $lr_{\text{base}} = 0.001$ and the decoder only $lr_{\text{base}} = 0.01$, the weight decay is set to 0.0001 and all images are cropped to $513 \times 513$ and $\mathcal{B}_l = \mathcal{B}_u = 3$.

For Cityscapes, all parameters use $lr_{\text{base}} = 0.01$, and the weight decay is set to 0.0005. The learning rate decay parameter is set to $\text{power} = 0.9$. Due to memory constraints, all images are cropped to $769 \times 769$ and $\mathcal{B}_\ell = \mathcal{B}_u = 2$. All experiments are conducted on a machine with 8 Nvidia RTX A5000 GPUs.

Figure E.1: **Example of refined pseudo-labels**, the structure is as in Fig. 3, the numbers under the predictions show the pixel-wise accuracy of the prediction map.

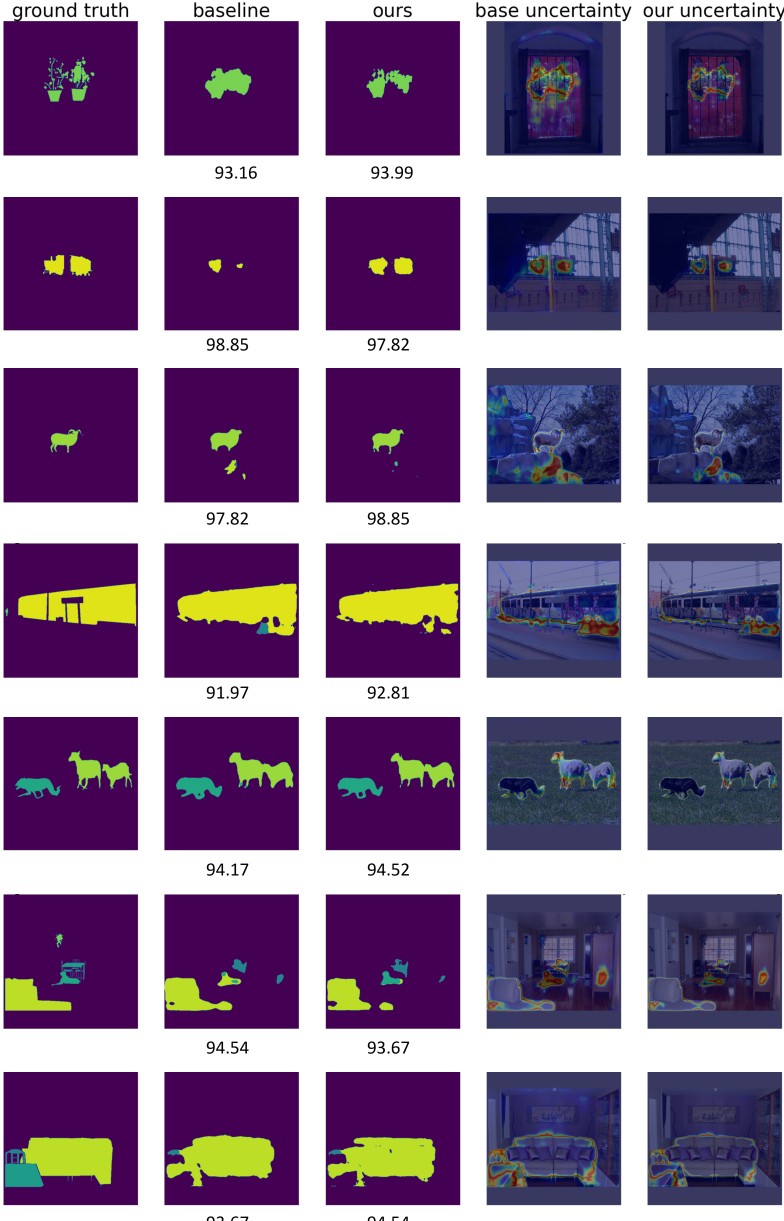

## E   More visual results

We present in Appendix E an extension of Fig. 3, showing more instances from the unlabeled data and the corresponding pseudo-labeled with the baseline model and S4MC.

Through our method, we can achieve more accurate predictions during the inference phase without any refinements. This results in the generation of more seamless and continuous predictions, which depict the spatial configuration of objects more accurately.