# OpenReview forum: "Semi-Supervised Semantic Segmentation via Marginal Contextual Information"
_NeurIPS.cc/2023/Conference — Submitted to NeurIPS 2023_

### Official Review · Reviewer_RDV7 · 2023-07-04

**Soundness:** 3 good
**Presentation:** 3 good
**Contribution:** 2 fair
**Rating:** 4
**Confidence:** 3

**Summary:**

The submission proposes a method to improve the quality of pseudo labels for semi-supervised semantic segmentation. The proposed approach leverages the spatial correlation of labels in segmentation maps by grouping neighboring pixels and considering their pseudo-labels, rather than generating pseudo labels at each pixel independently. Experiments show improvements over state-of-the-art semi-supervised semantic segmentation methods.

**Strengths:**

The proposed approach is conceptually simple and experimental results indicate it is effective.

**Weaknesses:**

1. It is not clear to me what is the improvement from considering neighbors when generating pseudo labels. It would be good if Table 4 (a) has an extra column with experiment using neighbor size 1x1, meaning no neighbor is used. Right now, the only comparison is with SOTA models that do not consider context, but the comparison is not convincing as there might be implementation differences. Also, why using a larger neighbor size hurts performance in Table 4 (a)?
2. The experiment setting is still a bit unclear to me. It would be great to state what images are used as labeled images and what images are unlabeled. Also, I think there misses an upper bounding experiment with the same model trained with all annotated images. Right now the SSL model is outperforming fully-supervised model, which does not make much sense to me.
3. Since Pascal VOC is a relatively small dataset, can we use more unlabeled images (e.g., COCO) to further improve performance?

**Questions:**

Please see weaknesses.

**Limitations:**

Please see weaknesses.

---

> ### Author Rebuttal · Authors · 2023-08-09
>
> **It is not clear to me what is the improvement from considering neighbors when generating pseudo labels. It would be good if Table 4 (a) has an extra column with experiment using neighbor size 1x1, meaning no neighbor is used. Right now, the only comparison is with SOTA models that do not consider context, but the comparison is not convincing as there might be implementation differences.**
>
> We would like to emphasize that in every experiment where a method is compared with and without S4MC, both are executed using an identical codebase. Therefore, the baseline inherently serves as the 1x1 neighborhood experiment that was requested.
> As for Table 4 (a), the experiment of 1x1 is reported in Table 1 (69.46 mIoU).
> The reproduction of the other baselines can be observed in Tables 1, 2, and 3 in the attached PDF.
> We will make this point more explicit in the revised manuscript for better clarity.
>
>
> **Why does using a larger neighbor size hurt performance in Table 4 (a)?**
>
>
> The degradation in refinement performance incurred by larger neighborhoods primarily stems from the diminished spatial coherence observable over greater distances.
>
> **The experiment setting is still a bit unclear to me. It would be great to state what images are used as labeled images and what images are unlabeled.
> Since Pascal VOC is a relatively small dataset, can we use more unlabeled images to further improve performance?**
>
>
> This work follows the standard setup in semi-supervised semantic segmentation, with various percentages of training samples used with labels, while the remaining samples are used as unlabeled. The same protocol was used in all methods in comparison.
> The first experiment uses Pascal VOC 12 dataset (up to 1464 images)  as labeled examples and the remaining examples as well as SBD[1] (9,118 images) as unlabeled examples.
>
> The second experiment uses one joint set from Pascal VOC 12, as well SBD (a total of 10,582 images), where the extra data from [1] is coarsely annotated.
>
> The third experiment uses the Cityscapes dataset (2976 images).
>
> While one may use COCO (or any other unlabeled images), this is the well-established baseline that allows us fair comparison in our experimental setup.
>
> Regarding selecting labeled images, our methodology aligns with the data split outlined in [1] for CutMix-Seg. Meanwhile, for FixMatch and Unimatch, we adhere to the data split detailed in [2]. A straightforward random data partitioning approach underpins experiments not covered within the references above. Notably, partial data splits can be accessed within our GitHub repository, with comprehensive details slated for inclusion upon the final publication of this manuscript.
>
>
> **I think there misses an upper bounding experiment with the same model trained with all annotated images. Right now the SSL model is outperforming the fully-supervised model, which does not make much sense to me.**
>
> We agree with the reviewer that we did not include the upper-bounding experiments in the paper.\
> The attainable upper performance bound (achieved under full supervision) is contingent on the underlying baseline methodology. In the Fixmatch+S4MC and Unimatch+S4MC, we recorded respective values of 81.65 and 80.15 for full supervision across PASCAL and Citiscapes, respectively. Similarly, for CutMix-Seg+S4MC, the figures for full supervision stood at 81.65 and 79.33 for the same benchmarks.
>
> If by outperforming fully-supervision, the reviewer intended that our results with $\frac{1}{2}$ of the annotated data surpass the results of other methods under the name “Full”, we hope that this rebuttal helps to clarify that this experiment utilizes fully supervision on PASCAL and no supervision on SBD. In the camera-ready version, we will make the experimental setting clear to the reader.
>
> [1] Semi-Supervised Semantic Segmentation Using Unreliable Pseudo-Labels, CVPR 2022
>
> [2] Revisiting Weak-to-Strong Consistency in Semi-Supervised Semantic Segmentation, CVPR 2023

---

> > ### Comment · Reviewer_RDV7 · 2023-08-22
> > **Response**
> >
> > Thanks to the authors for their response, the rebuttal addressed most of my concerns. However, I'm afraid I don't agree with "While one may use COCO (or any other unlabeled images), this is the well-established baseline that allows us fair comparison in our experimental setup."
> >
> > In my opinion, the goal of semi-supervised learning is to utilize as much unlabeled data as possible to improve performance and should not limit ourselves to the so-called "fair comparison". The standard setup is mainly for ablation and proper comparison, but on the other hand, we should show how the proposed method could be useful in practice by utilizing all available data, in order to make progress in the field of semi-supervised semantic segmentation.
> >
> > With this, I would take the answer as the proposed method may not work well with unlabeled COCO images and thus I would keep my original score.
> >
> > However, I'm okay with acceptance if all other reviewers including the AC think the proposed method could work well in large-scale setting.

---

> > > ### Author Response · Authors · 2023-08-22
> > > **Joint training with unlabeled COCO**
> > >
> > > Thank you for your insightful input.
> > >
> > >
> > > While your inquiry presents an intriguing configuration, establishing novel benchmarks falls outside the scope of the current study.

---

### Official Review · Reviewer_EHYQ · 2023-07-06

**Soundness:** 2 fair
**Presentation:** 2 fair
**Contribution:** 2 fair
**Rating:** 5
**Confidence:** 4

**Summary:**

In this paper, the authors propose a new confidence refinement scheme called S4MC for improving the pseudo-labels in semi-supervised semantic segmentation. Unlike existing methods, S4MC considers the spatial correlation of labels in segmentation maps by grouping neighboring pixels and collectively analyzing their pseudo-labels. This approach allows for the utilization of more unlabeled data during training while maintaining the quality of the pseudo-labels, all without significant computational overhead. Through extensive experiments on standard benchmarks, the authors demonstrate that S4MC surpasses current state-of-the-art methods in semi-supervised learning, offering a promising solution to reduce the cost of acquiring dense annotations.

**Strengths:**

+ The paper is written in a clear and concise manner, effectively presenting the motivation and ideas behind the proposed approach.
+ The proposed approach is rigorously evaluated through extensive experiments performed on multiple benchmarks.

**Weaknesses:**

- The contribution of this paper is somewhat limited, as the idea of leveraging contextual information from neighboring pixel predictions to enhance segmentation has been extensively explored in the literature.
- In Tables 1-3, it is evident that S4MC + FixMatch (Ours) demonstrates superior performance compared to S4MC + CutMix-Seg (Ours). However, the performance of FixMatch, U2PL+FixMatch, and PS-MT+FixMatch  is not explicitly mentioned in the given context.

**Questions:**

Please refer to the Weaknesses.

**Limitations:**

The authors have adequately addressed the limitations

---

> ### Author Rebuttal · Authors · 2023-08-09
>
> **The contribution of this paper is somewhat limited, as the idea of leveraging contextual information from neighboring pixel predictions to enhance segmentation has been extensively explored in the literature.**
>
> We agree with the reviewer's observation regarding the well-studied nature of contextual information from nearby pixels. While this concept has been explored in various contexts, our novelty lies in its adaptation to the task of semi-supervised learning. Specifically, we use spatial context for confidence refinement -- a mechanism specific to methods involving pseudo labels. To the best of our knowledge, this is the first work that has addressed this particular integration. Should the reviewer be aware of any relevant literature we may have overlooked, we would be grateful to delve into the specific distinctions during the discussion period.
>
>
> **In Tables 1-3, it is evident that S4MC + FixMatch (Ours) demonstrates superior performance compared to S4MC + CutMix-Seg (Ours). However, the performance of FixMatch, U2PL+FixMatch, and PS-MT+FixMatch is not explicitly mentioned in the given context.**
>
> Thank you for your valuable feedback. We appreciate your comment and want to emphasize that S4MC is highly adaptable and can seamlessly integrate with a wide range of existing semi-supervised learning methods. Our initial submission showcased its compatibility with FixMatch and CutMix-Seg, and based on reviewer insights, we have expanded our experiments to include the recent UniMatch technique [1]. We are pleased to report that S4MC consistently improved performance across nearly all configurations in our tests.
>
> To provide a more effective comparison of S4MC with other methods, we have included baseline performances of these three techniques in Tables 1, 2, and 3 in the attached PDF. We believe this enables a comprehensive overview of S4MC's potential to enhance the efficacy of various semi-supervised learning methods.
>
> [1] Revisiting Weak-to-Strong Consistency in Semi-Supervised Semantic Segmentation, CVPR2023

---

### Official Review · Reviewer_Utxo · 2023-07-07

**Soundness:** 3 good
**Presentation:** 3 good
**Contribution:** 3 good
**Rating:** 6
**Confidence:** 5

**Summary:**

This work focuses on semi-supervised semantic segmentation. To generate more accurate pseudo-labels, this work attempts to leverage the spatial correlation of labels in segmentation maps and proposes a confidence margin refinement module to model the contextual information. Extensive experiments have been conducted to validate the effectiveness of this method. The proposed method achieves promising performance on different datasets and settings.

**Strengths:**

Strengths:
The idea is interesting and the writing is clear.
The effectiveness of the proposed method has been validated on several standard benchmarks.


**Weaknesses:**

Weakness:
The time complexity of the proposed confidence margin refinement (CMR) module is necessary to be analyzed. According to L153, Eq9, and L168-170, the neighbor pixel selection will be performed for each class at every pixel location, that is the time complexity is proportional to the number of pixels, categories, and the number of unioned events. Since semantic segmentation is a dense prediction task, such a computation cost may not be neglected.
The example presented in L155-158 is not convincing to support the argument in L154. In this example, the author only considers one specific neighboring pixel which has a higher probability for the first class, but for the second class, it should search again with the Eq9 and that is not necessary to select the previous neighboring pixel but to select a pixel whose probability for the second class can most contribute to the interested uncertain pixel in a predefined neighborhood. Especially at edge pixels where the category changes, this kind of argument is hardly guaranteed.
L158-160 argues that the proposed CMR module can prevent the creation of over-confident predictions. However, the refined prediction \tilde{p}_c((x_{j,k})) = p_c(x_{j,k}) + p_c(x_{l,m}) (1 - p_c(x_{j,k})) based on Eq8 will definitely be greater than or at least equal to the original one p_c(x_{j,k}), that is the refined prediction will become more confident. It seems the argument is not reasonable.
The difference between S4MC+CutMix-seg and S4MC-FixMatch requires more explanation. Since the student-teacher modeling framework seems to be the default setting in this work and the CutMix operation is normally taken as a strong data augmentation in a semi-supervised semantic segmentation task[1][2], does that mean the latter method uses a stronger data augmentation to achieve a better performance? If it is, then what is the data augmentation?
The latest comparing methods is published in CVPR2022. Due to the rapid development of the semi-supervised semantic segmentation area, it is necessary to compare the proposed approach with more up-to-date methods, such as [1][2]. It seems that the proposed method is comparable to [1] and inferior to [2] in the CutMix setting.
[1] Semi-supervised Semantic Segmentation with Prototype-based Consistency Regularization, NeurIPS2022
[2] Revisiting Weak-to-Strong Consistency in Semi-Supervised Semantic Segmentation, CVPR2023


**Questions:**

Please see weakness above.

---

> ### Author Rebuttal · Authors · 2023-08-09
>
> We sincerely thank you for your insightful review. We believe that by implementing your suggested corrections, we can enhance the clarity, particularly in areas which may have initially seemed ambiguous.
>
> **The time complexity of the proposed confidence margin refinement (CMR) module is necessary to be analyzed, the neighbor pixel selection will be performed for each class at every pixel location. such a computation cost may not be neglected.**
>
> Thank you for raising this important point of discussion. We would like to emphasize that our refinement module's time complexity and memory footprint are low compared to backbone inference, despite the dense nature of our refinement operation.
> Let us denote the image size by $H,W$ and the number of classes by $C$.
>
> First, the predicted map of dimension $H \times W \times C$ is stacked with the padded-shifted versions, creating a tensor of shape [n,H,W,C]. K top neighbors are picked via top-k operation and calculate the union event as presented in Eq. (7).
> (pseudo code provided in pdf Alg 1).
>
> The overall space (memory) complexity of the calculation is O(n \times H \times W \times C), which is negligible considering all parameters and gradients of the model. Time complexity adds 3 tensor operations (stack, topk and multiplication) over the H \times W \times C tensor, where the multiplication operates k times, which means O(k \times H \times W \times C). This is again negligible for any reasonable number of classes compared to tens of convolutional layers with hundreds of channels.
>
> To verify that, we conducted a training time analysis comparing FixMatch and FixMatch + S4MC over PASCAL with 366 labeled examples, using distributed training with 8 Nvidia RTX 3090 GPUs.
> FixMatch average epoch takes 28:15 minutes, and FixMatch + S4MC average epoch takes 28:18 minutes, an increase of about 0.2% in runtime.
>
> **The example presented in L155-158 is not convincing to support the argument in L154. In this example, the author only considers one specific neighboring pixel which has a higher probability for the first class, but for the second class, it should search again with the Eq9.**
>
>
> This example relies on our observation that learning with few annotations suffers from low confidence on the surface of objects. It was intended to provide some background knowledge to the reader.
> However, we understand that it turned out more confusing than intuitive and did not contribute much to the method understanding. We will remove it in the camera-ready version.
>
>
>
> **L158-160 argues that the proposed CMR module can prevent the creation of over-confident predictions. However, the refined prediction based on Eq8 will be greater than or at least equal to the original one, the refined prediction will become more confident.**
>
> Note that our confidence metric is not the value of $\tilde{p}_c((x{j,k}))$. Instead, the metric is defined in Eq. (5): the difference between $\tilde{p}_c$ for the top two classes. Since this metric takes into account $\tilde{p}_c$ for the two most probable classes, the increase of $\tilde{p}_c$ for dominant class does not necessarily translates to a confidence increase. It is also important to take into account that $\tilde{p}_c$ does not define a probability distribution over classes, since it is not normalized, so comparison of $\tilde{p}_c$ and $p_c$ is not really indicative.
>
> The confidence increases if the gained information for the dominant class is larger than the gain of the second one. In Fig A1 (in the appendix), we analyze the contribution of our refinement and show that in the advanced stage of the learning, our refinement module is more effective in reducing overconfidence and, by that, reducing the false positive pseudo-labels.
>
> **The difference between S4MC+CutMix-seg and S4MC-FixMatch requires more explanation. CutMix operation is normally taken as a strong data augmentation in a SSL semantic segmentation task[1][2], does that mean the latter method uses a stronger data augmentation? If it is, then what is the data augmentation?**
>
> The observation is correct, as CutMix-Seg is based on a teacher-student regime with the same instance fed to both, and the teacher is EMA of the student weights, while FixMatch is based on image perturbation, extracting knowledge from weak to strong augmentation. In this paper, we follow the augmentations found in [1][2][3].
>
> For the Cutmix-seg baseline, we use resize, crop, horizontal flip, GaussianBlur, and Cutmix [4].
> For the FixMatch baseline, we use resize, crop, and horizontal flip for weak and strong augmentations and ColorJitter, RandomGrayscale, and Cutmix for strong augmentations.
>
> **The latest comparing methods is published in CVPR2022. It is necessary to compare the with more advanced methods.**
>
> Since the submission of the paper, we became aware of the mentioned publications, we added [5][6] to methods in comparison, as can be seen in pdf tables 1,2,3.
> Since our approach is independent of the SSL method, we were able to incorporate S4MC into the current state-of-the-art method (UniMatch), resulting in an improvement of up to 1.39 mIoU  (from 77.7% to 79.09%), further supporting our claims of the universality of S4MC.
>
>
> [1] Semi-supervised semantic segmentation needs strong, varied perturbations, BMVC 2020
>
> [2]Semi-Supervised Semantic Segmentation Using Unreliable Pseudo Labels, CVPR 2022.
>
> [3] ST++: Make Self-training Work Better for Semi-supervised Semantic Segmentation, CVPR 2022.
>
> [4] CutMix: Regularization Strategy to Train Strong Classifiers with Localizable Features, ICCV 2019
>
> [5] Semi-supervised Semantic Segmentation with Prototype-based Consistency Regularization, NeurIPS2022
>
> [6] Revisiting Weak-to-Strong Consistency in Semi-Supervised Semantic Segmentation, CVPR2023

---

> > ### Comment · Reviewer_Utxo · 2023-08-21
> >
> > I appreciate the reviewer's effort in answering my questions. Most of my concerns are addressed in the rebuttal and I am happy to raise my rating.

---

### Official Review · Reviewer_k6ru · 2023-07-17

**Soundness:** 3 good
**Presentation:** 3 good
**Contribution:** 3 good
**Rating:** 6
**Confidence:** 4

**Summary:**

To incorporate spatial contextual information in Semi-supervised segementation, the authors proposed S4MC, which refines confidence with the help of neighbors. This interesting refinement scheme aids in both adding falsely filtered pseudo-lables as well as removing erroeous ones. Additionally, a Dynamic Partition Adjustment module is employed to enable more pseduo-labeled pixels to be used. Results on several settings verified the effectiveness of S4MC.

**Strengths:**

* the confidence refinement scheme is novel and effective
* the evaluation is comprehensive and the results achieve new state-of-art
* the paper is well-written and the sturcture is clear
* easy to be followed

**Weaknesses:**

please refer to the questions listed below.

**Questions:**

* Some quantitative results should be provided to support the statement "refinement procedure aids in both adding falsely filtere pseduo-labels as well as removing erroneous ones.", instead of only qualitative analysis in Figure 3.
* Is the confidence refinement scheme independent of specific frameworks and can it be combined with other pseudo-label-based semi-supervised semantic segmentation methods (i.e., is it orthogonal to other methods)? No need for further experiments, just a small discussion here.
* What are the relationships between S4MC, CutMix-Seg, FixMatch, S4MC+CutMix-Seg and S4MC+FixMatch?
* It seems that the *DPA* is also an important component of the overall framework. However, since DPA is proposed by a previously published work. And the improvements brought by S4MC may come from both DPA and confidence refinement. Is it primarily driven by confidence refinement? Since confidence refinement is the most novel aspect of this paper. The authors should provide experimental results to demonstrate that the improvement brought by confidence refinement is the most significant part.

**Limitations:**

The authors should further discuss potential negative societal impact of this work and also should discuss the limitations of this work.

---

> ### Author Rebuttal · Authors · 2023-08-09
>
> We greatly appreciate your thoughtful review. The responses provided below include additional experiments and ablations, which are available in the attached PDF. We believe these additions will shed further light on aspects that may have been less clear in our previous explanations.
>
> **Some quantitative results should be provided to support the statement "refinement procedure aids in both adding falsely filtered pseudo-labels as well as removing erroneous ones.", instead of only qualitative analysis in Figure 3.**
>
>
> We quantify the effect of the refinement process by measuring the improvement in pseudo-labeling quantity and quality. Specifically, Fig. 4 shows that both the number of pseudo-labels being propagated for supervision (i.e., pass the threshold) and the accuracy of pseudo-labeling consistently improve throughout the training procedure when S4MC is applied. In Fig A1 (Appendix A), we further analyze the quality improvement by plotting true positive (TP) and false positive (FP) rates separately. From this plot, we can see both that TPR increases and FPR decreases.
>
> **Is the confidence refinement scheme independent of specific frameworks and can it be combined with other pseudo-label-based semi-supervised semantic segmentation methods (i.e., is it orthogonal to other methods)? No need for further experiments, just a small discussion here.**
>
> The refinement is indeed independent of a specific framework and is applicable to any framework that utilizes pseudo-labels for dense prediction under the spatial coherence assumption. We choose to demonstrate it using Cutmix-seg and FixMatch, as those are well-established baselines, as well as the most recent and strong method (Unimatch) [1]. Over those three baselines, we demonstrate a consistent improvement of our scheme.
> The improvements can be observed in pdf Tables 1,2 and 3.
>
>
> **What are the relationships between S4MC, CutMix-Seg, FixMatch, S4MC+CutMix-Seg and S4MC+FixMatch?**
>
> As correctly pointed out by the reviewer in the question above, S4MC is designed to enhance semi-supervision methods, such as CutMix-Seg and FixMatch. Thus, S4MC+X refers to the addition of SM4C to method X. CutMix-Seg uses the same augmentation for both teacher and student models that differ by EMA of each other,  while FixMatch uses weak augmentation strategy for the teacher and strong one for the student, where both share weights.
>
> We will ensure that this is clearly stated in the text and modify the comparison as shown in pdf Tables 1,2,3, where for each baseline method X, we provide the results of S4MC+X for direct comparison.
>
>
> **It seems that the DPA is also an important component of the overall framework. However, since DPA is proposed by a previously published work. And the improvements brought by S4MC may come from both DPA and confidence refinement. Is it primarily driven by confidence refinement? Since confidence refinement is the most novel aspect of this paper. The authors should provide experimental results to demonstrate that the improvement brought by confidence refinement is the most significant part.**
>
> Thank you for the suggestion. DPA is an important mechanism to set an appropriate  threshold to the state of the model at different training iterations, as proposed in [2]. Based on the reviewers' suggestion, we added an ablation on the use of DPA in pdf table 4.
>
> S4MC achieves improved performance with and without DPA (1.87% and 1.09%, respectively). Conversely, we find that DPA provides improvement (0.18%) only when used with S4MC and reduces performance by 0.6%.
>
>
> **The authors should further discuss the potential negative societal impact of this work, and also should discuss the limitations of this work.**
>
> We express our gratitude to the reviewer for highlighting this significant concern.
>
> Societal impact:
>
> Similar to the majority of semi-supervised models, we utilize a small subset of annotated data, which has the potential to introduce biases from the data into the model. Further, our PLR module assumes spatial coherence. While that holds for natural images, it may yield adverse effects in other domains, such as medical imaging. It is important to consider these potential impacts before choosing to use our proposed method.
>
>
> Limitations:
>
> The constraint imposed by the spatial coherence assumption also restricts the applicability of this work to dense prediction tasks. Improving the quality of pseudo-labels for overarching tasks such as classification might necessitate reliance on data distribution and the exploitation of inter-sample relationships. We currently explore this avenue of research.
>
> We are committed to incorporating these impacts and limitations into the final version.
>
>
> [1] Revisiting Weak-to-Strong Consistency in Semi-Supervised Semantic Segmentation, CVPR2023
>
> [2] Semi-Supervised Semantic Segmentation Using Unreliable Pseudo Labels, CVPR 2022.

---

> > ### Comment · Reviewer_k6ru · 2023-08-21
> >
> > Thanks the authors for the response; my concerns have been resolved. I still believe that this work is effective and novel, and I will maintain my rating.

---

### Author Rebuttal · Authors · 2023-08-09

We thank the reviewers for their diligent assessment of our manuscript.

We appreciate your recognition of our contribution's novelty and effectiveness (k6ru). Furthermore, we are pleased that you found both the clarity and comprehension of our work to be satisfactory (k6ru, utxom, ehyq, rdv7). The comprehensive evaluations and impactful results have also been positively acknowledged (k6ru, utxom, ehyq, rdv7).

We have diligently taken into account the concerns and inquiries highlighted in this review. We believe that these insights helped us significantly improve the overall quality of our submission, in particular the presentation of the proposed method.

We will implement the essential revisions outlined in the comments for the camera-ready version. These modifications include the addition of a 1x1 column to the ablation study in Table 4 (a), removing the confusing example, incorporating a limitation section, and including the additional results presented in the attached PDF. Furthermore, we remain attentive to any additional comments that the reviewers may raise during the discussion period.


The attached PDF includes Tables 1, 2, and 3, which expand comparison with prior art and new experiments conducted of Unimatch + S4MC for all reported datasets and partitions.
Table 4 presents an ablation study of the components of our approach.
Algorithm 1 introduces a PyTorch-like pseudo-code for our pseudo-label refinement procedure.
It is important to emphasize that S4MC can be seamlessly integrated into a multitude of existing SSL segmentation frameworks, consistently yielding improvements across the various frameworks we assessed.


UniMatch, a robust recent approach, employs a combination of three students per supervision signal, one feature-level augmentation (FP), and two potent augmentations (S1 and S2). While our fusion with this approach did not yield statistically significant enhancements, this outcome prompted us to evaluate the distinct contributions of UniMatch's components thoroughly. We adopted the pixel agreement measurement described in Eq. (10) to assess the impact, which involves all eight neighboring pixels. Remarkably, our findings reveal that the feature perturbation branch affects pixel agreement similar to that of S4MC.
The visual representation provided in Figure 1 showcases the distribution of spatial agreement using FixMatch (S1), DusPerb (S1, S2), UniMatch (S1, S2, FP), and the amalgamation of DusPerb with S4MC (S1, S2). Furthermore, Figure 2 presents the distribution of this measurement throughout training.

We hope the further details provided in this response allow for more comprehensive consideration of our contribution and would appreciate it if reviewers who believe their concerns were addressed revise their score accordingly.

---

### Decision · Program_Chairs · 2023-09-21

**Decision:**

Reject

**Comment:**

This paper proposes to use contextual information for semi-superivsed segmentation. In particular, the approach considers the spatial correlation of labels in segmentation maps by grouping neighboring pixels and analyzing their pseudo-labels. Experimental results on PASCAL and Cityscapes demonstrate the effectiveness. Concerns were raised regarding the complexity of the approach, missing comparisons with more recent literatures, limited novelty and datasets used.

The authors made good effort in the rebuttal and clarified some technical details and added more results. The reviewers still have concerns regarding the scalabitity of the proposed approach as most experiments are conducted on PASCAL.  While it is fine to use PASCAL, semi-supervised segmentation mainly aims to improve results by leveraging a substantial amount of unlabeled samples. It would make the paper more convincing if more large-scale benchmarks are used. Based on the review and the discussion, the AC recommends rejection, and encourages the authors to consider the comments for future submission.